# *Banana*: *Bana*ch Fixed-Point *N*etwork for Pointcloud Segmentation with Inter-Part Equiv*a*riance

**Congyue Deng**[1]* **Jiahui Lei**[2]* **Bokui Shen**[1] **Kostas Daniilidis**[2,3] **Leonidas Guibas**[1]
[1] Stanford University [2] University of Pennsylvania [3] Archimedes, Athena RC
{congyue, willshen, guibas}@cs.stanford.edu, {leijh, kostas}@cis.upenn.edu

## Abstract

Equivariance has gained strong interest as a desirable network property that inherently ensures robust generalization. However, when dealing with complex systems such as articulated objects or multi-object scenes, effectively capturing inter-part transformations poses a challenge, as it becomes entangled with the overall structure and local transformations. The interdependence of part assignment and per-part group action necessitates a novel equivariance formulation that allows for their co-evolution. In this paper, we present *Banana*, a Banach fixed-point network for pointcloud segmentation with inter-part equivariance **by construction**. Our key insight is to iteratively solve a fixed-point problem, where point-part assignment labels and per-part $SE(3)$-equivariance co-evolve simultaneously. We provide theoretical derivations of both per-step equivariance and global convergence, which induces an equivariant final convergent state. Our formulation naturally provides a strict definition of inter-part equivariance that generalizes to unseen inter-part configurations. Through experiments conducted on both articulated objects and multi-object scans, we demonstrate the efficacy of our approach in achieving strong generalization under inter-part transformations, even when confronted with substantial changes in pointcloud geometry and topology.

## 1 Introduction

From articulated objects to multi-object scenes, multi-body systems, i.e. shapes composed of multiple parts where each part can be moved separately, are prevalent in various daily-life scenarios. However, modeling such systems presents considerable challenges compared to rigid objects, primarily due to the infinite shape variations resulting from inter-part pose transformations with exponentially growing complexities. Successfully modeling these shapes demands generalization across potential inter-part configurations. While standard data augmentation techniques can potentially alleviate such problems, exhaustive augmentation can be highly expensive especially given the complexity of such systems compared to rigid objects. Meanwhile, recent rigid-shape analysis techniques have made significant progress in building generalization through the important concept of equivariance [39, 14, 79, 2, 81, 71, 20, 11, 15, 3, 36, 42, 55]. Equivariance dictates that when a transformation is applied to the input data, the network's output should undergo a corresponding transformation. For example, if a is rotated by 90 degrees, the segmentation masks should also rotate accordingly. However, to the best of our knowledge, no existing work has managed to extend the formulation of equivariance to inter-part configurations.

In this paper, we address the challenge of achieving inter-part equivariance in the context of part segmentation, a fundamental task in analysis, which is also the key to generalizing equivaraince from single object to multi-body system. Extending equivariance to multi-body systems presents notable challenges in both formulation and realization. When dealing with a shape that allows for inter-part motions, the model must exhibit equivariance to both global and local state changes, which can only be defined by combining part assignment and per-part transformations. For example, to model an

37th Conference on Neural Information Processing Systems (NeurIPS 2023).

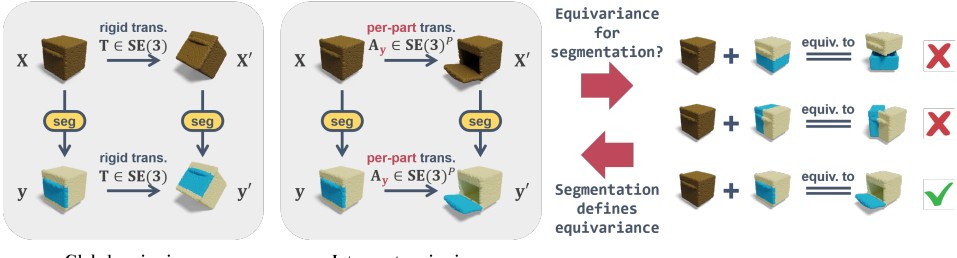

Figure 1: **"Chicken or the egg" problem with inter-part equivariance. Left:** Rigid $SE(3)$-equivariance. If a rigid transformation is applied to the input, the output segmentation transforms accordingly. **Right:** Inter-part equivariance. Analogously, we want the input and output to be coherent under *per-part* transformations. But such a definition of equivariance requires part segmentations, which is also exactly the desired network output, resulting in a "chicken or the egg" problem.

oven's inter-part states, we first need to differentiate between the parts corresponding to the door and the body and then define the movement of each individual part. An intriguing "chicken or the egg" problem arises when attempting to build inter-part equivariance without access to a provided segmentation, where segmentation is necessary to define equivariance, but segmentation itself is the desired output that we aim to produce.

Our key insight to tackle this seeming dilemma is to model inter-part equivariance as a sequential fixed-point problem by co-evolving part segmentation labels and per-part $SE(3)$-equivariance. We provide theoretical derivations for both the per-step behavior and global convergence behavior, which are crucial aspects of a fixed-point problem. We show that our formulation establishes per-step equivariance through network construction, which then induces an overall inter-part equivariance upon convergence. Thus, by having equivariant per-step progression and global convergence, our formulation naturally gives rise to a strict definition of inter-part equivariance through iterative inference that generalizes to unseen part configurations. We further bring our formulation to concrete model designs by proposing a novel part-aware equivariant network with a weighted message-passing paradigm. With localized network operators and per-part $SE(3)$-equivaraint features, the network is able to guarantee per-step inter-part equivariance as well as facilitate stable convergence during the iterative inference. We test our framework on articulated objects to generalize from static rest states to novel articulates and multi-object scenes to generalize from clean synthetic scenes to cluttered real scans. Our model shows strong generalization in both scenarios even under significant changes in pointcloud geometry or topology.

To summarize, our key contributions are

- To the best of our knowledge, we are the first to provide a strict definition of inter-part equivariance for pointcloud segmentation and introduce a learning framework with such equivariance **by construction**.
- We propose a fixed-point framework with one-step training and iterative inference and show that the per-step equivariance induces an overall equivariance upon convergence.
- We design a part-aware equivariant message-passing network with stable convergence.
- Experiments show our strong generalization under inter-part configuration changes even when they cause subsequent changes in pointcloud geometry or topology.

## 2   Related Work

**Equivariant pointcloud networks**. Existing works in equivariant 3D learning mainly focus on rigid $SE(3)$ transformations. As a well-studied problem, it has developed comprehensive theories [39, 14, 79, 2, 81] and abundant network designs [71, 20, 11, 15, 3, 36, 42, 55], which benefit a variety of 3D vision and robotics tasks ranging from pose estimation [44, 46, 54, 61, 89], shape reconstruction [11, 10], to object interaction and manipulation [19, 27, 60, 63, 80, 82]. A few recent works have extended $SE(3)$ equivariance to part level [86, 41, 50]. [86] employs a local equivariant feature extractor for object bounding box prediction, showing robustness under object-level and scene-level pose changes. [41] learns an $SE(3)$-equivaraint object prior and applies it to object

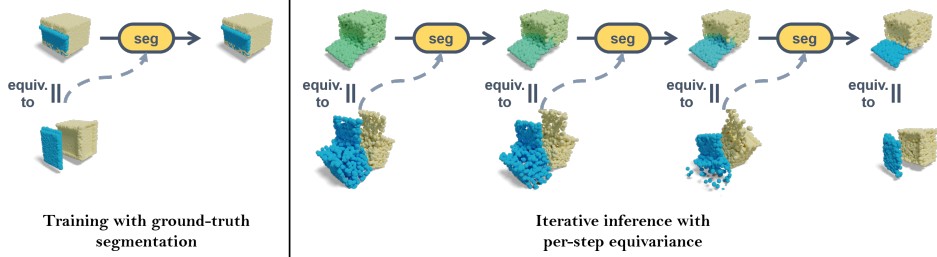

Figure 2: **Single-state training and novel-state iterative inference. Left:** Training with ground-truth segmentation as both network input and output. **Right:** Equivariant Banach iterations. At each step $k$, the network is equivariant to its current segmentation input $\mathbf{y}^{(k)}$ by construction. At the end of the iterations, it converges to its fixed point with an induced overall equivariance.

detection and segmentation in scenes with robustness to scene configuration changes. [50] learns part-level equivariance and pose canonicalization from a collection of articulated objects with a two-stage coarse-to-fine network. However, all these works are purely heuristics-based, and none has provided a strict definition of inter-part equivariance.

**Multi-body systems**. From small-scale articulated objects to large-scale multi-object scenes, there is a wide range of 3D observations that involve multiple movable parts or objects, exhibiting various configurations and temporal variations. This has sparked a rich variety of works dedicated to addressing the challenges in multi-body systems, studying their part correspondences [56, 49, 8, 48, 43], segmentations [85, 76, 67, 31, 30, 70, 7, 12], reconstructions [34, 37, 32, 53, 40], or rearrangements [78, 68], to name just a few. However, as we know the system is acted via products of group actions; a more structured network that exploits such motion prior by construction is desired. Only a few works have studied this problem. [86] exploits global and local gauge equivariance in 3D object detection via bottom-up point grouping but without a systematical study of inter-object transformations. [41] further exploits equivariance of object compositions in scene segmentation, but as an EM-based approach, it requires exhaustive and inefficient enumerations and no global convergence is guaranteed. In contrast, our approach offers a theoretically sound framework that achieves multibody equivariance by construction, which enables robust analysis for various tasks and provides a more structured and reliable solution compared to existing methods.

**Iterative inference**. From recurrent neural networks [23, 29] to flow [16, 59, 17, 38, 83] or diffusion-based generative models [65, 66, 28, 64, 51, 88], iterative inference has played many diverse yet essential roles in the development of computer vision. Though naturally adapted to sequential or temporal data, iterative inference can also be applied to static learning problems such as image analysis [22, 21, 73, 26] and pointcloud segmentation [84, 47]. Recent studies also show that iterative inference presents stronger generalizability than one-step forward predictions [62, 6], with explanations of their relations to the "working memory" of human minds [4, 5] or human visual systems [45, 35]. In our work, we employ iterative inference on static pointclouds to co-evolve two intertwined attributes: part segmentation and inter-part equivaraince, in order to address a seemingly contradictory situation.

## 3 Inter-Part Equivariance

We start by defining inter-part equivariance on pointclouds with given part segmentations (Sec. 3.1). Then, we will introduce how to extend this definition to unsegmented pointclouds using a fixed-point framework (Sec. 3.2) with one-step training and iterative inference. The iterations have per-step equivariance by network construction, which is shown to induce an overall equivariance upon convergence (Sec 3.3). Finally, we will also explain how to eliminate part orders for instance segmentation (Sec 3.4).

### 3.1 Multi-Body Systems

Let $\mathbf{X} \in \mathbb{R}^{N \times 3}$ be a pointcloud with $P$ parts. Segmentations on $\mathbf{X}$ are represented as point-part assignment matrices $\mathbf{y} \in [0, 1]^{N \times P}$ which sum up to 1 along the $P$ dimension. As the action of $\mathrm{SE}(3)^P$ on $\mathbf{X}$ and its resulting equivariance is subject to the part decompositions, we define the action of $\mathrm{SE}(3)^P$ on (pointcloud, segmentation) pairs $(\mathbf{X}, \mathbf{y})$. When $\mathbf{y}$ is a binary mask, the $P$

parts of $\mathbf{X}$ are $P$ disjoint sub-pointclouds and each $\mathrm{SE}(3)$ component of $\mathrm{SE}(3)^P$ acts on one part separately. More concretely, for any transformation $\mathbf{A} = (\mathbf{T}_1, \cdots, \mathbf{T}_P) \in \mathrm{SE}(3)^P$, we define $\mathbf{A}(\mathbf{X}, \mathbf{y}) := (\mathbf{X}', \mathbf{y})$ where the $n$-th point $\mathbf{x}_n$ is transformed to $\mathbf{x}'_n$ by

$$\mathbf{x}'_n := \sum_{p=1}^{P} \mathbf{y}_{np}(\mathbf{x}_n \mathbf{R}_p + \mathbf{t}_p). \tag{1}$$

where $\mathbf{R}_p$ and $\mathbf{t}_p$ are the rotation and translation components of $\mathbf{T}_p \in \mathrm{SE}(3)$. For soft segmentation masks, we view each point as probabilistically assigned to one of the $P$ parts.

**Equivariance**. Similar to $\mathrm{SE}(3)$-equivariance on rigid shapes $f(\mathbf{T}\mathbf{X}) = \mathbf{T}f(\mathbf{X}), \forall \mathbf{T} \in \mathrm{SE}(3)$, we define the inter-part equivariance on $(\mathbf{X}, \mathbf{y})$ pairs by saying that a function $f$ is $\mathrm{SE}(3)^P$-equivariant if $\forall \mathbf{A} \in \mathrm{SE}(3)^P$,

$$f(\mathbf{A}(\mathbf{X}, \mathbf{y})) = \mathbf{A}f(\mathbf{X}, \mathbf{y}). \tag{2}$$

Note that if the function outputs $f(\mathbf{X}, \mathbf{y})$ are no longer (pointcloud, segmentation) pairs, the $\mathrm{SE}(3)^P$-action on its outputs needs to be specified. A special case is when $\mathrm{SE}(3)^P$ acts trivially on $f(\mathbf{X}, \mathbf{y})$ by $f(\mathbf{A}(\mathbf{X}, \mathbf{y})) = f(\mathbf{X}, \mathbf{y})$, which defines the common-sense "invariance". For a pointcloud segmentation network with per-point part-label outputs $\mathbf{y} \in [0, 1]^{N \times P}$, we desire it to be $\mathrm{SE}(3)^P$-invariant under the above definitions, that is $f(\mathbf{A}(\mathbf{X}, \mathbf{y})) \equiv \mathbf{y}, \forall \mathbf{A} \in \mathrm{SE}(3)^P$.

### 3.2 Banach Fixed-Point Iterations

**"Chicken or the egg"**. The "chicken or the egg" nature of segmentation with inter-part equivariance immediately emerges from the above definition: to enforce $\mathrm{SE}(3)^P$-equivariance for a neural network $f(\cdot; \Theta)$, the part segmentation $\mathbf{y}$ is required as input, which is also exactly the desired output of the network (Fig. 1). That this,

$$f(\mathbf{X}, \mathbf{y}; \Theta) = \mathbf{y}. \tag{3}$$

To resolve this dilemmatic problem, we approach Eq. 3 as a fixed-point equation on function $f(\cdot, \mathbf{y}; \Theta)$ *w.r.t.* $\mathbf{y}$. And instead of using single-step forward predictions, we solve it with Banach fixed-point iterations (Fig. 2).

**Training**. At training time, we aim to optimize the network weights $\Theta$ such that the labeled s $(\mathbf{X}_i, \mathbf{y}_i), i \in \mathcal{I}$ from the dataset become fixed points of the function $f(\cdot, \Theta)$:

$$\Theta_* = \arg\min_{\Theta} \frac{1}{|\mathcal{I}|} \sum_{i \in \mathcal{I}} \|f(\mathbf{X}_i, \mathbf{y}_i; \Theta) - \mathbf{y}_i\|. \tag{4}$$

However, there exists a trivial solution which is the identity function $f(\cdot, \mathbf{y}; \Theta) \equiv \mathbf{y}$, meaning that any arbitrary segmentation can satisfy the objective for any pointclouds. Thus, to avoid such degenerated cases, we have to limit the network expressivity. More specifically, we limit the Lipschitz constant $L$ of the network *w.r.t.* $\mathbf{y}$ which is defined by

$$\|f(\mathbf{X}, \mathbf{y}_1; \Theta) - f(\mathbf{X}, \mathbf{y}_2; \Theta)\| \leq L\|\mathbf{y}_1 - \mathbf{y}_2\|, \quad \forall \mathbf{X}, \forall \mathbf{y}_1, \mathbf{y}_2. \tag{5}$$

If $f$ degenerates to the identical mapping, its Lipschitz $L = 1$, which violates the above constraint. When $L < 1$, $f$ is a contractive function and has a unique fixed point based on the Banach fixed-point theorem. Our training objective is to align this fixed point with the ground-truth segmentation.

**Iterative inference**. At inference time, given an input pointcloud $\mathbf{X}_0$, the learnable parameters $\Theta_*$ fixed in $f$ and we look for a segmentation $\mathbf{y}$ that satisfies

$$f_{\mathbf{X}_0, \Theta_*}(\mathbf{y}) := f(\mathbf{X}_0, \mathbf{y}; \Theta_*) = \mathbf{y}. \tag{6}$$

We solve this equation with Banach fixed-point iterations [1]

$$\mathbf{y}^* = \lim_{k \to \infty} \mathbf{y}^{(k)} = \lim_{k \to \infty} f^{(k)}_{\mathbf{X}_0, \Theta_*}(\mathbf{y}^{(0)}) \tag{7}$$

where $\mathbf{y}^{(0)}$ is a random initialization of the segmentation on $\mathbf{X}_0$. For a contractive function $f$, the uniqueness of $\mathbf{y}^*$ induces a well-defined mapping from pointclouds to segmentations $f_{\Theta}^* : \mathbf{X}_0 \to \mathbf{y}^*$.

### 3.3 Equivariance

For an input pairs $(\mathbf{X}, \mathbf{y})$ with known segmentation $\mathbf{y}$, we employ an SE(3)-equivariant backbone [15] on each individual part and construct an $\text{SE}(3)^P$-equivariant network, which will be further explained in Section 4. This guarantees the inter-part equivariance at training time and at each timestep during inference, but most importantly we desire the overall $\text{SE}(3)^P$-equivariance at the convergence point of the Banach iteration. Here we prove two properties:

**Self-coherence of convergent state**. At each inference timestep $k$, we have $\text{SE}(3)^P$-equivariance subject to $\mathbf{y}^{(k)}$, that is, $\forall \mathbf{A} \in \text{SE}(3)^P$,

$$f(\mathbf{A}(\mathbf{X}_0, \mathbf{y}^{(k)}); \Theta) = f(\mathbf{X}_0, \mathbf{y}^{(k)}; \Theta), \quad \forall k \in \mathbb{N}, \tag{8}$$

Because of the continuity of $f$ and the compactness of $[0, 1]^{N \times P}$, we can take limits *w.r.t.* $k$ on both sides, which gives

$$f(\mathbf{A}(\mathbf{X}_0, \mathbf{y}^*); \Theta) = f(\mathbf{X}_0, \mathbf{y}^*; \Theta) = f_\Theta^*(\mathbf{X}_0). \tag{9}$$

This shows the self-coherence of the convergent point between $f_\Theta^*$ and $\mathbf{y}^*$.

**Generalization to novel inter-part states**. Suppose the network has seen $(\mathbf{X}, \mathbf{y}_{\text{gt}})$ in the training set and learned $f_\Theta(\mathbf{X}, \mathbf{y}_{\text{gt}}; \Theta_*) = \mathbf{y}_{\text{gt}}$ with the loss function in Eq. 4 minimized to zero. Now apply an inter-part transformation $\mathbf{A} \in \text{SE}(3)^P$ and test the network on $(\mathbf{X}', \mathbf{y}_{\text{gt}}) := \mathbf{A}(\mathbf{X}, \mathbf{y}_{\text{gt}})$. We would like to show $f_\Theta^*(\mathbf{X}') = f_\Theta^*(\mathbf{X}) = \mathbf{y}_{\text{gt}}$. First of all, we know that $\mathbf{y}_{\text{gt}}$ itself is a fixed point of $f(\mathbf{X}', \cdot; \Theta)$, which is due to the equivariance *w.r.t.* $\mathbf{y}_{\text{gt}}$ at training time:

$$f(\mathbf{X}', \mathbf{y}_{\text{gt}}; \Theta) = f(\mathbf{A}(\mathbf{X}, \mathbf{y}_{\text{gt}}); \Theta) = f(\mathbf{X}, \mathbf{y}_{\text{gt}}; \Theta). \tag{10}$$

Now if $f$ is a contractive function on $\mathbf{y}$, we have the **uniqueness** of Banach fixed-points and thus $\mathbf{y}_{\text{gt}}$ is **the only** fixed point for $f(\mathbf{X}', \cdot; \Theta)$, implying that the iterations in Eq. 7 must converge to $\mathbf{y}_{\text{gt}}$. Equivariance of the per-step iteration and the final convergent state are illustrated in Fig. 2 right.

In less ideal cases when the training loss is not zero but a small error $\varepsilon$, we view it as the distance $\|\mathbf{y}^{(1)} - \mathbf{y}^{(0)}\|$ between the ground-truth $\mathbf{y}^{(0)} = \mathbf{y}_{\text{gt}}$ and the one-step iteration output $\mathbf{y}^{(1)}$. The distance between the actual fixed-point $\mathbf{y}^*$ and the ground-truth $\mathbf{y}_{\text{gt}}$ is then bounded by

$$\|\mathbf{y}^* - \mathbf{y}_{\text{gt}}\| = \sum_{k=0}^{\infty} \|\mathbf{y}^{(k+1)} - \mathbf{y}^{(k)}\| \leq \sum_{k=0}^{\infty} L^k \|\mathbf{y}^{(1)} - \mathbf{y}^{(0)}\| = \frac{1}{1-L}\varepsilon \tag{11}$$

where $L$ is the Lipschitz constant of the network.

### 3.4 Part Permutations

A pointcloud segmentation represented by $\mathbf{y} \in [0, 1]^{N \times P}$ inherently carries an order between the $P$ parts, which is the assumption in semantic segmentation problems. But in various real-world scenarios, it is common to encounter situations where multiple disjoint parts are associated with the same semantic label, without any clear or coherent part orderings. In order to tackle this challenge, we also extend our method to encompass the instance segmentation problem without part orderings. For simplicity, here we assume that all parts can be permuted together by $\mathfrak{S}_P$. In practice, permutations only occur among the parts within each semantic label, for which we can easily substitute the $\mathfrak{S}_P$ below with its subgroup and the conclusion still holds. We define an equivalence relation on $[0, 1]^{N \times P}$ by

$$\mathbf{y}_1 \sim \mathbf{y}_2 \iff \exists \sigma \in \mathfrak{S}_P, s.t. \; \mathbf{y}_1\sigma = \mathbf{y}_2, \tag{12}$$

which gives us a quotient space $[0, 1]^{N \times P}/\mathfrak{S}_P$ and each equivalent class $\hat{\mathbf{y}}$ in this quotient space represents an instance segmentation without part ordering. We can define a metric on $\hat{\mathbf{y}}$ by

$$d(\hat{\mathbf{y}}_1, \hat{\mathbf{y}}_2) := \min_{\sigma \in \mathfrak{S}_P} \|\mathbf{y}_1\sigma - \mathbf{y}_2\|, \tag{13}$$

which makes $\left([0, 1]^{N \times P}/\mathfrak{S}_P, d\right)$ a complete metric space, on which the Banach fixed-point theorem holds and so are the iterations and convergence properties. Proofs for the well-definedness of $d$ and the completeness of the quotient space are shown in the supplementary material.

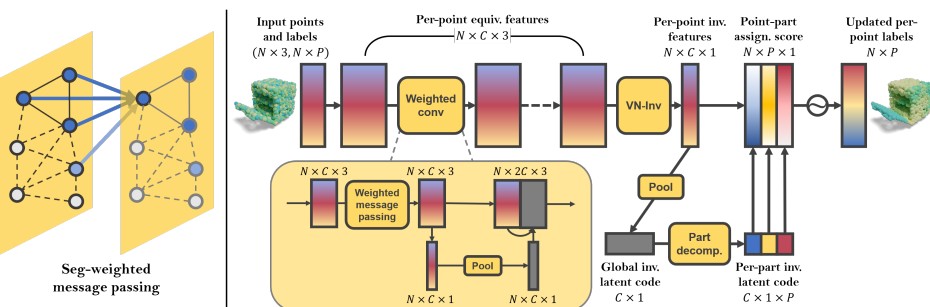

Figure 3: **Part-aware equivariant network. Left:** Segmentation-weighted message passing. **Right:** Overall architecture for segmentation label update. Color gradients on the tensors indicate the usage of segmentation labels,

## 4 Part-Aware Equivariant Network

In this section, we present our part-aware equivariant network for segmentation label updates (Fig.3). We utilize a SE(3)-equivariant framework called Vector Neurons (VN) [15] as the backbone to encode per-point features $\mathbf{V} \in \mathbb{R}^{N \times C \times 3}$, ensuring equivariance within each part. Subsequently, these features are transformed into invariant representations, eliminating relative poses to facilitate inter-part information propagation and global aggregations.

**Segmentation-weighted message passing**. Key to our part-aware equivariant network is a message-passing module weighted by the input segmentation $\mathbf{y}$ (Fig. 3 left). Intuitively, given a point $\mathbf{x}_n \in \mathbf{X}$ with latent feature $\mathbf{V}_n$, for its neighborhood points $\mathbf{x}_m \in \mathcal{N}$, we only allow information propagation between $\mathbf{x}_n$ and $\mathbf{x}_m$ if they belong to the same part. When the part segmentation $\mathbf{y}$ is a soft mask, we compute the probability $p_{nm}$ of $\mathbf{x}_n$ and $\mathbf{x}_m$ belonging to the same part by $p_{nm} = \mathbf{y}_n \mathbf{y}_m^t$, $\mathbf{y}_n, \mathbf{y}_m \in [0,1]^{1 \times P}$ and weight the local message passing with

$$\mathbf{V}'_n = \sum_{m \in \mathcal{N}} p_{nm}\, \varphi(\mathbf{V}_m - \mathbf{V}_n, \mathbf{V}_n) \Big/ \sum_{m \in \mathcal{N}} p_{nm}. \tag{14}$$

Proof of inter-part equivariance for this module can be found in the supplementary material.

**Network architecture**. Fig. 3 right shows the overall network architecture for segmentation label updates. Given an input pair $(\mathbf{X}, \mathbf{y})$, we first extract a per-point SE(3)-equivariant local feature $\mathbf{V} \in \mathbb{R}^{N \times C \times 3}$ within each part masked by $\mathbf{y}$. $\mathbf{V}$ is then passed through a sequence of weighted convolutions with the massage passing defined in Eq. 14. To enable efficient inter-part information exchange, after each message-passing layer, we compute a global invariant feature through a per-point VN-invariant layer, followed by a global pooling and a concatenation between per-point and global features. After the point convolutions, the per-point equivariant features $\mathbf{V}_n$ are converted to invariant features $\mathbf{S}_n$ with inter-part poses canceled. We then apply a global pooling on $\mathbf{S}_n$ to obtain a global shape code, which is decomposed into $P$ part codes $\mathbf{Q}_1, \cdots, \mathbf{Q}_P$ as in [13]. Finally, we compute a point-part assignment score for all $(\mathbf{S}_n, \mathbf{Q}_p)$ pairs (resulting in an $N \times P$ tensor) and apply a softmax activation along its $P$ dimension to obtain the updated segmentation $\mathbf{y}'$. For instance segmentation without part permutations (Sec. 3.4), we modify the computation of the part feature $\mathbf{Q}$ by replacing the global feature decomposition with point feature grouping based on $\mathbf{y}$.

For pointcloud networks with set-invariance across all points, directly restricting the upper bound of the network Lipschitz using weight truncations [72, 24, 87] will greatly harm network expressivity as it limits the output range on each point. On the other hand, the space of all possible segmentations $[0,1]^{N \times P}$ has extremely high dimensionality, making the computation of Lipschitz regularization losses [69] rather inefficient. Therefore, we do not explicitly constrain the Lipschitz constant of our network. Nevertheless, we confine all operations involving $\mathbf{y}$ to local neighborhoods per point, as shown in Eq. 14. This practical approach generally ensures a small Lipschitz constant for the network in most scenarios. The stability of fixed-point convergence will be studied in Sec. 5.3.

## 5 Experiments

### 5.1 Articulated-object part segmentation

We begin by evaluating our method on articulated-object part segmentation using on four categories of the Shape2Motion dataset [74]: washing machine, oven, eyeglasses, and refrigerator. To demon-

| Setting | Unseen states | | | | Unseen states + unseen instances | | | |
|---|---|---|---|---|---|---|---|---|
| Category | Washing machine | Oven | Eye-glasses | Refrige-rator | Washing machine | Oven | Eye-glasses | Refrige-rator |
| PointNet [57] | $46.18_{\pm3.84}$ | $44.08_{\pm8.97}$ | $38.96_{\pm18.41}$ | $38.37_{\pm12.32}$ | $46.15_{\pm3.18}$ | $45.29_{\pm9.54}$ | $39.21_{\pm17.14}$ | $39.00_{\pm12.13}$ |
| DGCNN [75] | $46.78_{\pm4.37}$ | $44.30_{\pm10.84}$ | $33.35_{\pm20.96}$ | $39.70_{\pm14.05}$ | $46.60_{\pm4.03}$ | $46.69_{\pm12.76}$ | $34.96_{\pm22.14}$ | $40.13_{\pm13.81}$ |
| VNN [15] | $47.35_{\pm1.93}$ | $53.64_{\pm13.27}$ | $57.08_{\pm15.52}$ | $52.36_{\pm11.39}$ | $47.09_{\pm1.84}$ | $51.01_{\pm11.09}$ | $62.98_{\pm11.09}$ | $48.49_{\pm11.58}$ |
| VNN-Inv | $46.58_{\pm1.61}$ | $62.32_{\pm14.49}$ | $37.24_{\pm7.76}$ | $60.78_{\pm15.57}$ | $46.66_{\pm1.15}$ | $67.35_{\pm16.50}$ | $38.19_{\pm10.26}$ | $63.18_{\pm14.98}$ |
| **Ours** | $\mathbf{82.32}_{\pm15.08}$ | $\mathbf{81.91}_{\pm10.81}$ | $\mathbf{77.78}_{\pm14.45}$ | $\mathbf{77.26}_{\pm7.79}$ | $\mathbf{84.99}_{\pm11.76}$ | $\mathbf{82.84}_{\pm8.13}$ | $\mathbf{78.65}_{\pm10.36}$ | $\mathbf{73.93}_{\pm14.08}$ |

Table 1: **Shape2Motion results.** Numbers are segmentation IoU multiplied by 100.

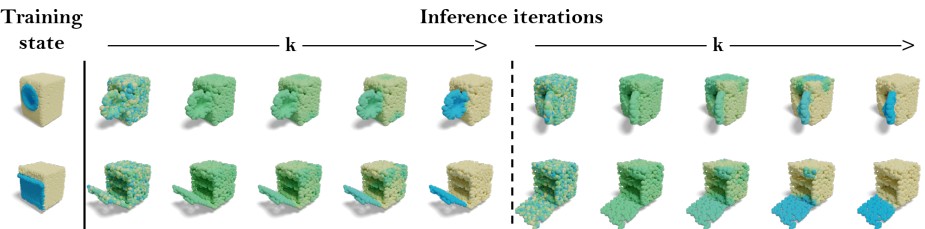

Figure 4: **Banach iterations on Shape2Motion.** The network is trained on rest-state objects (left) and tested on novel articulation states.

strate the generalizability of our approach, we train our model on objects in a single rest state, such as ovens with closed doors (as depicted in Fig. 5, left). This training setup aligns with many synthetic datasets featuring static shapes [9, 52]. Subsequently, we assess the performance of our model on articulated states, where global and inter-part pose transformations are applied, replicating real-world scenarios. We evaluate our model on both unseen articulation states of the training instances (Tab. 1, left) and on unseen instances (Tab. 1, right). In both cases, the joint angles are uniformly sampled from the motion range for each category. We compare our network to the most widely adopted object-level part segmentation networks PointNet [57] and DGCNN [75] without equivariance, plus VNN [15] with global SE(3)-equivariance. We also compare to a VNN variant, named VNN-Inv, which first computes per-point local SE(3)-equivariant features and then converts them to invariant features and applies invariant message passing. This is a strategy adopted by [18] for human segmentation. All IoUs are computed for semantic parts. In the case of eyeglasses and refrigerators, which have two parts sharing the same semantic label, we train our network with three motion part inputs but output two semantic part labels. This is feasible because our weighted message passing (Eq. 14) is agnostic to the number of parts.

Fig. 4 demonstrates our iterative inference process on novel states after training on the rest states. Fig. 5 shows our segmentation predictions on the four categories. The inter-part pose transformations can cause significant changes in pointcloud geometry and topology, yet our method can robustly generalize to the unseen articulations with inter-part equivariance.

## 5.2 Multi-Object Scans

We also test our instance segmentation framework (without part orders) on multi-object scans.

**Segmentation transfer on DynLab**. DynLab [30] is a collection of scanned laboratory scenes, each with 2-3 rigidly moving solid objects captured under 8 different configurations with object positions randomly changed. We overfit our model to the first configuration of each scene and apply it to the 7 novel configurations, transfering the segmentation from the first scan to the others via inter-object equivariance.

We compare our method to a variety of multi-body segmentation methods, including motion-based co-segmentation methods (DeepPart [85], NPP [25], MultiBodySync [30]), direct segmentation prediction models (PointNet++ [58], MeteorNet [49]), geometry-based point grouping (Ward-linkage [77]), and pre-trained indoor instance segmentation modules (PointGroup [33]). The baseline setups follow [30]. Tab. 2 shows the segmentation IoUs. One thing to note is that motion-based co-segmentation can also be viewed as learning inter-object equivariance via Siamese training, yet they cannot reach the performance of equivariance by construction.

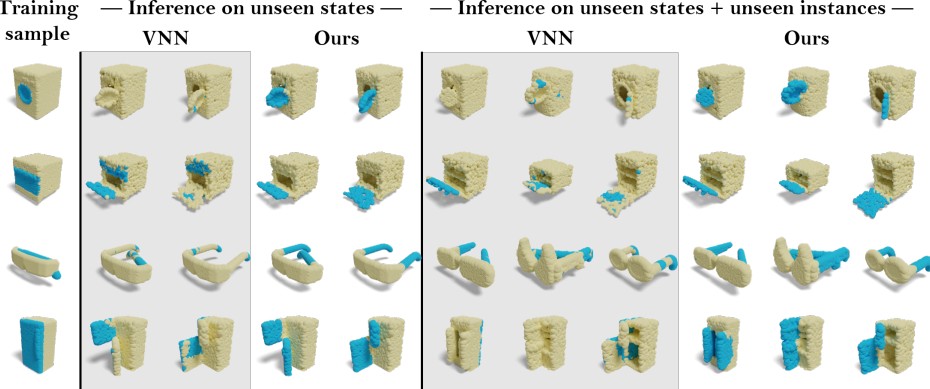

| Training sample | — Inference on unseen states — | | — Inference on unseen states + unseen instances — | |
| | VNN | Ours | VNN | Ours |

Figure 5: **Shape2Motion part segmentation results.** *Global poses are aligned here for visualization but we do not assume it in our inference.*

| Method | Seg. IoU |
|---|---|
| PointNet++ [58] | $39.4_{\pm 7.1}$ |
| MeteorNet [49] | $71.8_{\pm 9.7}$ |
| DeepPart [85] | $66.3_{\pm 17.2}$ |
| NPP [25] | $71.6_{\pm 7.7}$ |
| Ward-linkage [77] | $88.6_{\pm 5.8}$ |
| PointGroup [33] | $72.4_{\pm 12.5}$ |
| MultiBodySync [30] | $94.0_{\pm 3.1}$ |
| **Ours** | $\mathbf{95.5}_{\pm 4.6}$ |

Table 2: **DynLab segmentation.**

Figure 6: **Chair scan segmentation.**

**Synthetic to real chair scans**. We train our model using a synthetic dataset constructed from clean ShapeNet chair models [9] with all instances lined up and facing the same direction (Fig. 6 left). We then test our model on the real chair scans from [41] with diverse scene configurations (Fig. 6 right). The configurations range from easy to hard, including: **Z** (all chairs standing on the floor), **SO(3)** (chairs laid down), and **Pile** (chairs piled into clutters). Remarkably, our model with inter-object equivariance demonstrates successful generalization across all these scenarios, even in the most challenging Pile setting with cluttered objects.

### 5.3 Ablation Studies

**Network operators and convergence**. We show the importance of our locally confined message passing and $SE(3)$-features (Eq. 14) to the convergence of test-time iterations. As opposed to extracting part-level equivariant features, one can also apply pose canonicalizations to each part individually and employ non-equivariant learning methods in the canonical space. For $SE(3)$-transformations with a translation component in $\mathbb{R}^3$ and a rotation component in $SO(3)$, the former can be canonicalized by subtracting the part centers and the latter by per-part PCA. Thus we compare our full model with two ablated versions with *part-level* equivariance replaced by canonicalization: **PCA**, where we canonicalize both the part translations and rotations and simply employ a non-equivariant DGCNN [75] for the weighted message passing; **VNN**, where we only canonicalize the part translations but preserve the $SO(3)$-equivariant structures in the network.

Tab. 3 shows the segmentation IoUs of the full model and the ablated versions on the Shape2Motion oven category. espite canonicalizations being agnostic to per-part $SE(3)$ transformations, they rely on absolute poses and positions, which breaks the locality of network operations, resulting in a significant increase in the Lipschitz constant of the network. Consequently, when any local equivariant operator is replaced by canonicalization, the network performance experiences a drastic drop.

To further examine the convergence ranges, we test the three models under different $\mathbf{y}^{(0)}$ initializations by adding random noises $\xi \sim \mathcal{U}(0,1)$ to $\mathbf{y}_{\text{gt}}$ according to $\mathbf{y}^{(0)} = (1-\alpha)\mathbf{y}_{\text{gt}} + \alpha\xi$, $\alpha \in [0,1]$. Fig. 7 shows the performance changes of the three models with gradually increased noise levels

| Method | Equiv. | Pose cano. | Seg. IoU |
|--------|--------|-----------|----------|
| PCA | – | rot.+trans. | $31.22_{\pm 7.65}$ |
| VNN | SO(3) | trans. | $27.86_{\pm 13.07}$ |
| **Ours** | SE(3) | – | $\mathbf{82.84}_{\pm 8.13}$ |

Table 3: **Ablations of per-part equivariant operators replaced by pose canonicalization.** Numbers are IoU multiplied by 100.

Figure 7: **Model performances with initializations of different noise levels.**

| Lipschitz constraint | Weight trunc. [72] | Reg. loss (rand.) | Reg. loss (rand.) | Reg. loss (adv.) [69] | None |
|----------------------|--------------------|--------------------|--------------------|------------------------|------|
| Norm | $l_\infty$ | $l_\infty$ | $l_2$ | $l_2$ | - |
| IoU ($\times 10^2$) | $43.99_{\pm 2.43}$ | $31.32_{\pm 21.56}$ | $71.29_{\pm 15.97}$ | $43.99_{\pm 2.43}$ | $80.68_{\pm 10.96}$ |

Table 4: **Network performances under different Lipschitz constraints.**

on $\mathbf{y}^{(0)}$. The PCA rotation canonicalization has a theoretically unbounded Lipschitz constant, making the network unable to converge stably even within a small neighborhood of $\mathbf{y}_{\text{gt}}$. The SO(3)-equivariant VNN with translation canonicalization perfectly overfits to the ground-truth fixed point but exhibits a rapid decline in performance as the initial $\mathbf{y}^{(0)}$ deviates from $\mathbf{y}_{\text{gt}}$. In contrast, our SE(3)-equivariant network demonstrates stable performance across different noise levels.

**Lipschitz constraints..** We also provide a study of different Lipschitz constraining methods under different norms in Tab. 4. For weight truncation [72], we use the $l_\infty$-Lipschitz as it is the loosest constraint and set the per-layer Lipschitz upper bound to be 0.99999. For regularization losses, we set the overall network Lipschitz threshold to 0.99. The adversarial sampling [69] loss only works with $l_2$-norm. We reduce the total number of points from 2048 to 1024 and the GNN neighbors from 40 to 20 due to the large memory consumption for the loss computations. Based on our observations, not having an explicit Lipschitz constraint and using the SE(3)-equivariant message passing work best for the current situation, yet none of these existing Lipschitz constraining methods are helpful to the network performance. We also plot the $l_2$-norm regularization losses in the supplementary material, showing that they are zero almost everywhere.

## 6 Conclusions

In this work, we propose *Banana*, which provides both theoretical insights and experimental analysis of inter-part equivariance. While equivariance is typically examined from an algebraic perspective, we approach it from an analysis standpoint and obtain a strict formulation of inter-part equivariance at the convergent point of an iteration sequence. Based on our theoretical formulation, we propose a novel framework that co-evolves segmentation labels and per-part SE(3)-equivariance, showing strong generalization under both global and per-part transformations even when these transformations result in subsequent changes to the pointcloud geometry or topology. Experimentally, we show that our model can generalize from rest-state articulated objects to unseen articulations, and from synthetic toy multi-object scenes to real scans with diverse object configurations.

**Limitations and future work**. While our local segmentation-weighted message passing reduces the Lipschitz constant of the network, practically providing stability during test-time iterations, it is important to note that it is not explicitly bounded or regularized. Lipschitz bounds and regularizations for set-invariant networks without compromising network expressivity would be an interesting and important problem for future study. In addition, we study the full $\text{SE}(3)^P$ action on multi-body systems, but in many real-world scenarios, different parts or objects may not move independently due to physical constraints. Limiting the $\text{SE}(3)^P$-action to its subset of physically plausible motions can potentially increase our feature-embedding conciseness and learning efficiency.

**Broader Impacts**. Our study focuses on the general 3D geometry and its equivariance properties. Specifically, we investigate everyday appliances in common scenes, for which we do not anticipate any direct negative social impact. However, we acknowledge that our work could potentially be misused for harmful purposes. On the other hand, our research also has potential applications in areas that can benefit society, such as parsing proteins in medical science and building home robots for senior care.

## Acknowledgments and Disclosure of Funding

We gratefully acknowledge the following grants: a TRI University 2.0 grant, a Vannevar Bush Faculty Fellowship, and a gift from the Adobe Corporation awarded to Stanford University; and NSF FRR 2220868, NSF IIS-RI 2212433, NSF TRIPODS 1934960, NSF CPS 2038873 awarded to the University of Pennsylvania.

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
