# *Banana*: *Bana*ch Fixed-Point *N*etwork for Pointcloud Segmentation with Inter-Part Equiv*a*riance
## – Supplementary Material –

**Congyue Deng**[1]* **Jiahui Lei**[2]* **Bokui Shen**[1] **Kostas Daniilidis**[2,3] **Leonidas Guibas**[1]
[1] Stanford University [2] University of Pennsylvania [3] Archimedes, Athena RC
{congyue, willshen, guibas}@cs.stanford.edu, {leijh, kostas}@cis.upenn.edu

This supplementary document provides proof for the statements in the main paper, as well as network and training details including the compute. We also provide a supplemental video for a quick introduction to our method and additional results. We will also make our code public upon paper acceptance.

## S.1 Proof for Part Permutations (Sec 3.4)

In this section, we show that $[0,1]^{N \times P}/\mathfrak{S}_P$ is a complete metric space with metric

$$d(\hat{\mathbf{y}}_1, \hat{\mathbf{y}}_2) := \min_{\sigma \in \mathfrak{S}_P} \|\mathbf{y}_1\sigma - \mathbf{y}_2\|.$$

### S.1.1 Well-Definednes of $d$

As $d$ is defined on equivalent classes, we need to show the well-definedness of $d$ in that $d$ is independent of the choice of representatives. Suppose $\mathbf{y}_1 = \mathbf{y}'_1$, $\mathbf{y}_2 = \mathbf{y}'_2$, then by the definition of equivalent classes, $\exists \sigma_1, \sigma_2 \in \mathfrak{S}_P$, *s.t.* $\mathbf{y}_1\sigma_1 = \mathbf{y}'_1$, $\mathbf{y}_2\sigma_2 = \mathbf{y}'_2$. On one side,

$$d(\hat{\mathbf{y}}_1, \hat{\mathbf{y}}_2) = \min_{\sigma \in \mathfrak{S}_P} \|\mathbf{y}_1\sigma - \mathbf{y}_2\| \le \|\mathbf{y}_1\sigma_1\sigma_2^{-1} - \mathbf{y}_2\|$$
$$= \|\mathbf{y}_1\sigma_1 - \mathbf{y}_2\sigma_2\| = d(\hat{\mathbf{y}}'_1, \hat{\mathbf{y}}'_2).$$

Similarly, we also have $d(\hat{\mathbf{y}}'_1, \hat{\mathbf{y}}'_2) \le d(\hat{\mathbf{y}}_1, \hat{\mathbf{y}}_2)$, and thus $d(\hat{\mathbf{y}}_1, \hat{\mathbf{y}}_2) = d(\hat{\mathbf{y}}'_1, \hat{\mathbf{y}}'_2)$.

### S.1.2 $d$ as a Metric

To show that $d$ is a metric, we need to show its **positivity**, **symmetry**, and **triangle inequality**.

**Positivity**. Suppose $\hat{\mathbf{y}}_1 \ne \hat{\mathbf{y}}_2$. Then $\forall \sigma \in \mathfrak{S}_P, \mathbf{y}_1\sigma \ne \mathbf{y}_2$. This implies

$$d(\hat{\mathbf{y}}_1, \hat{\mathbf{y}}_2) = \min_{\sigma \in \mathfrak{S}_P} \|\mathbf{y}_1\sigma - \mathbf{y}_2\| > 0$$

as $\mathfrak{S}_P$ is a finite set and taking minimum over it strictly preserves inequality. Similarly, we can show that $d(\hat{\mathbf{y}}, \hat{\mathbf{y}}) \ge 0$. Together with

$$d(\hat{\mathbf{y}}, \hat{\mathbf{y}}) = \min_{\sigma \in \mathfrak{S}_P} \|\mathbf{y}\sigma - \mathbf{y}\| \le \|\mathbf{y}\mathbf{1} - \mathbf{y}\| = \|\mathbf{y} - \mathbf{y}\| = 0,$$

we get $d(\hat{\mathbf{y}}, \hat{\mathbf{y}}) = 0$.

37th Conference on Neural Information Processing Systems (NeurIPS 2023).

**Symmetry**. For any $\hat{\mathbf{y}}_1, \hat{\mathbf{y}}_2$, we have

$$
\begin{aligned}
d(\hat{\mathbf{y}}_1, \hat{\mathbf{y}}_2) &= \min_{\sigma \in \mathfrak{S}_P} \|\mathbf{y}_1\sigma - \mathbf{y}_2\| \\
&= \min_{\sigma \in \mathfrak{S}_P} \|\mathbf{y}_1\sigma\sigma^{-1} - \mathbf{y}_2\sigma^{-1}\| \\
&= \min_{\sigma \in \mathfrak{S}_P} \|\mathbf{y}_1 - \mathbf{y}_2\sigma^{-1}\| \\
&= \min_{\sigma \in \mathfrak{S}_P} \|\mathbf{y}_2\sigma^{-1} - \mathbf{y}_1\| \\
&= \min_{\sigma' \in \mathfrak{S}_P} \|\mathbf{y}_2\sigma' - \mathbf{y}_1\| = d(\hat{\mathbf{y}}_2, \hat{\mathbf{y}}_1).
\end{aligned}
$$

Here we use the symmetry of the standard Euclidean norm $\|\cdot\|$, the $l_2$-distance perseverance of permutation matrices, and the fact that $\sigma \in \mathfrak{S}_P \iff \sigma^{-1} \in \mathfrak{S}_P$ due to the group structure of $\mathfrak{S}_P$.

**Triangle inequality**.

$$
\begin{aligned}
d(\hat{\mathbf{y}}_1, \hat{\mathbf{y}}_3) &= \min_{\sigma \in \mathfrak{S}_P} \|\mathbf{y}_1\sigma - \mathbf{y}_3\| = \min_{\sigma \in \mathfrak{S}_P} \|\mathbf{y}_1\sigma - \mathbf{y}_2\sigma' + \mathbf{y}_2\sigma' - \mathbf{y}_3\|, \quad \forall \sigma' \in \mathfrak{S}_P \\
&= \min_{\sigma' \in \mathfrak{S}_P} \min_{\sigma \in \mathfrak{S}_P} \|\mathbf{y}_1\sigma - \mathbf{y}_2\sigma' + \mathbf{y}_2\sigma' - \mathbf{y}_3\| \\
&\leq \min_{\sigma',\sigma \in \mathfrak{S}_P} \|\mathbf{y}_1\sigma - \mathbf{y}_2\sigma'\| + \|\mathbf{y}_2\sigma' - \mathbf{y}_3\| \\
&= \min_{\sigma',\sigma \in \mathfrak{S}_P} \|\mathbf{y}_1\sigma\sigma'^{-1} - \mathbf{y}_2\sigma'\sigma'^{-1}\| + \|\mathbf{y}_2\sigma' - \mathbf{y}_3\| \\
&= \min_{\sigma',\sigma'' \in \mathfrak{S}_P} \|\mathbf{y}_1\sigma'' - \mathbf{y}_2\| + \|\mathbf{y}_2\sigma' - \mathbf{y}_3\| \\
&= \min_{\sigma'' \in \mathfrak{S}_P} \|\mathbf{y}_1\sigma'' - \mathbf{y}_2\| + \min_{\sigma'' \in \mathfrak{S}_P} \|\mathbf{y}_2\sigma' - \mathbf{y}_3\| \\
&= d(\hat{\mathbf{y}}_1, \hat{\mathbf{y}}_2) + d(\hat{\mathbf{y}}_2, \hat{\mathbf{y}}_3).
\end{aligned}
$$

### S.1.3 Completeness of $\left([0,1]^{N \times P} / \mathfrak{S}_P, d\right)$

To show the completeness of this space, we need to show that every Cauchy sequence converges to a point in this space. Suppose $\{\hat{\mathbf{y}}_i : i \in \mathbb{N}\}$ is a Cauchy sequence satisfying $\forall \varepsilon > 0, \exists N > 0, s.t. \forall i, j > M, d(\hat{\mathbf{y}}_i, \hat{\mathbf{y}}_j) < \varepsilon$. Let $\{\mathbf{y}_i : i \in \mathbb{N}\}$ be a sequence of arbitrarily selected representatives from each $\hat{\mathbf{y}}_i$, it is a bounded sequence in $[0,1]^{N \times P}$, and thus have a convergent subsequence $\{\mathbf{y}_{i_k} : k \in \mathbb{N}\}$ with limit $\lim_{k \to \infty} \mathbf{y}_{i_k} = \mathbf{y}^* \in [0,1]^{N \times P}$ by the Bolzano–Weierstrass theorem.

We would like to show that $\hat{\mathbf{y}}^* \in [0,1]^{N \times P} / \mathfrak{S}_P$ is the limit point of $\{\hat{\mathbf{y}}_i\}$ using proof by contradiction. Suppose it is not, then $\exists \varepsilon_0 > 0, s.t. \forall M > 0, \exists j_0 > M, s.t. d(\hat{\mathbf{y}}_{j_0}, \hat{\mathbf{y}}^*) \geq \varepsilon_0$. By the convergence of $\{\mathbf{y}_{i_k}\}$, $\exists M_1, s.t. \forall i_k \geq k > M_1, \|\mathbf{y}_{i_k} - \mathbf{y}^*\| < \varepsilon_0/3$. On the other hand, because $\{\hat{\mathbf{y}}_i\}$ is Cauchy, $\exists M_2, s.t. , \forall i, j > M_2, d(\hat{\mathbf{y}}_i, \hat{\mathbf{y}}_j) < \varepsilon_0/3$. Now let $M = \max(M_1, M_2)$, then for $i_k, j_0 > M$, we have

$$
\varepsilon_0 \leq d(\hat{\mathbf{y}}_{j_0}, \hat{\mathbf{y}}^*) \leq d(\hat{\mathbf{y}}_{j_0}, \hat{\mathbf{y}}_{i_k}) + d(\hat{\mathbf{y}}_{i_k}, \hat{\mathbf{y}}^*) < \varepsilon_0/3 + \|\mathbf{y}_{i_k} - \mathbf{y}^*\| < \varepsilon/3 + \varepsilon/3.
$$

A contradiction!

## S.2 Proof for Network Equivariance (Sec. 4)

Suppose $\mathbf{y}$ is binary, when $\mathbf{x}_n, \mathbf{x}_m$ belong to different parts $(p(m) \neq p(n))$, $\sum_{p \in P} \mathbf{y}_{np}\mathbf{y}_{mp} = 0$. The weighted message passing can be written as

$$
\begin{aligned}
\mathbf{V}'_n &= \sum_{m \in \mathcal{N}} \mathbf{y}_n \mathbf{y}_m^t \, \varphi(\mathbf{V}_m - \mathbf{V}_n, \mathbf{V}_n) \Big/ \sum_{m \in \mathcal{N}} \mathbf{y}_n \mathbf{y}_m^t \\
&= \sum_{m \in \mathcal{N}} \sum_{p \in P} \mathbf{y}_{np}\mathbf{y}_{mp} \, \varphi(\mathbf{V}_m - \mathbf{V}_n, \mathbf{V}_n) \Big/ \sum_{m \in \mathcal{N}} \mathbf{y}_n \mathbf{y}_m^t \\
&= \sum_{p(m)=p(n)} \varphi(\mathbf{V}_m - \mathbf{V}_n, \mathbf{V}_n) \Big/ \sum_{m \in \mathcal{N}} \mathbf{y}_n \mathbf{y}_m^t.
\end{aligned}
$$

Now apply a transformation $\mathbf{A} = (\mathbf{T}_1, \cdots, \mathbf{T}_P) \in \mathrm{SE}(3)^P$ to the input per-point features $\mathbf{V}$, it transforms $\mathbf{V}_n, \mathbf{V}_m$ into

$$\mathbf{V}_n \mapsto \sum_{p=1}^{P} \mathbf{y}_{np}(\mathbf{V}_n \mathbf{R}_p + \mathbf{t}_p), \quad \mathbf{V}_m \mapsto \sum_{p=1}^{P} \mathbf{y}_{mp}(\mathbf{V}_m \mathbf{R}_p + \mathbf{t}_p).$$

When $p(m) = p(n)$, $\sum_{p=1}^{P}(\mathbf{y}_{mp} - \mathbf{y}_{np}) = 0$, and the right-hand side of the above equation then turns into

$$\sum_{p(m)=p(n)} \varphi\bigg( \sum_{p=1}^{P} \mathbf{y}_{mp}(\mathbf{V}_m \mathbf{R}_p + \mathbf{t}_p) - \sum_{p=1}^{P} \mathbf{y}_{np}(\mathbf{V}_n \mathbf{R}_p + \mathbf{t}_p), \sum_{p=1}^{P} \mathbf{y}_{np}(\mathbf{V}_n \mathbf{R}_p + \mathbf{t}_p) \bigg) \bigg/ \sum_{m \in \mathcal{N}} \mathbf{y}_n \mathbf{y}_m^t$$

$$= \sum_{p(m)=p(n)} \varphi\bigg( \sum_{p=1}^{P} \mathbf{y}_{np}((\mathbf{V}_m - \mathbf{V}_n)\mathbf{R}_p + \mathbf{t}_p) + \sum_{p=1}^{P}(\mathbf{y}_{mp} - \mathbf{y}_{np})(\mathbf{V}_m \mathbf{R}_p + \mathbf{t}_p),$$

$$\sum_{p=1}^{P} \mathbf{y}_{np}(\mathbf{V}_n \mathbf{R}_p + \mathbf{t}_p) \bigg) \bigg/ \sum_{m \in \mathcal{N}} \mathbf{y}_n \mathbf{y}_m^t$$

$$= \sum_{p(m)=p(n)} \varphi\bigg( \sum_{p=1}^{P} \mathbf{y}_{np}((\mathbf{V}_m - \mathbf{V}_n)\mathbf{R}_p + \mathbf{t}_p), \sum_{p=1}^{P} \mathbf{y}_{np}(\mathbf{V}_n \mathbf{R}_p + \mathbf{t}_p) \bigg) \bigg/ \sum_{m \in \mathcal{N}} \mathbf{y}_n \mathbf{y}_m^t$$

$$= \sum_{p(m)=p(n)} \varphi\bigg( (\mathbf{V}_m - \mathbf{V}_n)\mathbf{R}_{p(n)} + \mathbf{t}_{p(n)}, \mathbf{V}_n \mathbf{R}_{p(n)} + \mathbf{t}_{p(n)} \bigg) \bigg/ \sum_{m \in \mathcal{N}} \mathbf{y}_n \mathbf{y}_m^t.$$

By the part-wise equivariance of $\varphi$, this is equal to $\mathbf{V}_n' \mathbf{R}_{p(n)} + \mathbf{t}_{p(n)}$.

## S.3 Network Lipschitz (Sec. 4)

Here we provide an (informal) explanation of why our $\mathrm{SE}(3)$-equivariant message passing architecture is helpful to a small network Lipschitz. We consider a metric on per-point segmentation labels and features defined by $d(\mathbf{y}, \mathbf{y}') := \max_n \|\mathbf{y}_n - \mathbf{y}_n'\|_2$, $d(\mathbf{V}, \mathbf{V}') := \max_n \|\mathbf{V}_n - \mathbf{V}_n'\|_2$. This is a mixture of $l_\infty$-norm across points and $l_2$-norm for per-point features. Intuitively, the $l_\infty$-norm is for dimensionality-independent properties, where the Lipschitz constant of the network won't drastically increase as the number of points increases.

For a message-passing layer $f(\mathbf{V}, \mathbf{y})$ as defined in Eq. (14), denote the edge operator between pairs of adjacent points $\varphi_{nm} = \varphi(\mathbf{V}_m - \mathbf{V}_n, \mathbf{V}_m)$, and its difference of function outputs with inputs $(\mathbf{V}, \mathbf{y})$ and $(\mathbf{V}', \mathbf{y}')$ can be upper-bounded by

$$d(f(\mathbf{V}, \mathbf{y}), f(\mathbf{V}', \mathbf{y}'))$$

$$= \max_n \bigg\| \sum_{m \in \mathcal{N}} \frac{p_{nm}\varphi_{nm}}{\sum_{m \in \mathcal{N}} p_{nm}} - \sum_{m \in \mathcal{N}} \frac{p_{nm}'\varphi_{nm}'}{\sum_{m \in \mathcal{N}} p_{nm}'} \bigg\|$$

$$= \max_n \bigg( \bigg\| \sum_{m \in \mathcal{N}} \frac{p_{nm}\varphi_{nm}}{\sum_{m \in \mathcal{N}} p_{nm}} - \sum_{m \in \mathcal{N}} \frac{p_{nm}\varphi_{nm}'}{\sum_{m \in \mathcal{N}} p_{nm}} \bigg\| + \bigg\| \sum_{m \in \mathcal{N}} \frac{p_{nm}\varphi_{nm}'}{\sum_{m \in \mathcal{N}} p_{nm}} - \sum_{m \in \mathcal{N}} \frac{p_{nm}'\varphi_{nm}'}{\sum_{m \in \mathcal{N}} p_{nm}'} \bigg\| \bigg)$$

$$\leq \max_n \bigg( \bigg\| \sum_{m \in \mathcal{N}} \frac{p_{nm}}{\sum_{m \in \mathcal{N}} p_{nm}} \bigg\| \cdot \|\varphi_{nm} - \varphi_{nm}'\| + \sum_{m \in \mathcal{N}} \|\varphi_{nm}'\| \cdot \bigg\| \frac{p_{nm}}{\sum_{m \in \mathcal{N}} p_{nm}} - \frac{p_{nm}'}{\sum_{m \in \mathcal{N}} p_{nm}'} \bigg\| \bigg)$$

As both $\sup \big\| \sum_{m \in \mathcal{N}} \frac{p_{nm}}{\sum_{m \in \mathcal{N}} p_{nm}} \big\|$ and $\sup \sum_{m \in \mathcal{N}} \|\varphi_{nm}'\|$ can be upper-bounded by constants, bounding $\|\varphi_{nm} - \varphi_{nm}'\|$ and $\big\| \frac{p_{nm}}{\sum_{m \in \mathcal{N}} p_{nm}} - \frac{p_{nm}'}{\sum_{m \in \mathcal{N}} p_{nm}'} \big\|$ w.r.t. $d(\mathbf{y}, \mathbf{y}')$ and $d(\mathbf{V}, \mathbf{V}')$ are key to the Lipschitz bound of $f(\mathbf{V}, \mathbf{y})$.

For network architectures with per-point operations such as PointNet, the second term is 0 as $p_{nm} = p_{nm}' \equiv \mathbf{1}_{\{n=m\}}$ and the first term degenerates to $\varphi_{nm} = \varphi(\mathbf{V}_n)$. In such a case, limiting the

Lipschitz of the layer is equivalent to limiting $\|\varphi(\mathbf{V}_n) - \varphi(\mathbf{V}_m)\|$ *w.r.t.* $d(\mathbf{V}, \mathbf{V}')$, which is directly limiting the Lipschitz of the per-point operation $\varphi$ and will affect the output range of the network.

For a message-passing network, the second term means the change of layer output caused by the change of part assignments in the neighborhood $\mathcal{N}$, which is bounded by $\sup \|\varphi_{nm}\|$ and the size of $\mathcal{N}$. The ablation study in the main paper focuses on the first term

$$\|\varphi(\mathbf{V}_m - \mathbf{V}_n, \mathbf{V}_m) - \varphi(\mathbf{V}'_m - \mathbf{V}'_n, \mathbf{V}'_m).\|$$

For our SE(3)-equivariant message passing with local features, all computations in $\varphi$ directly operates on $\mathbf{V}_n, \mathbf{V}_m$. But for canonicalization-based methods, geometric transformations (translations and/or rotations) are first applied to $\mathbf{V}_n, \mathbf{V}_m$ based on the input part assignment $\mathbf{y}$, which introduce excessive Lipschitz to $\varphi$ *w.r.t.* $\mathbf{y}$.

## S.4 Network and Training Details

In our network, we use 2 weighted message-passing layers to extract the per-point SE(3)-equivariant features, followed by 4 weighted message-passing layers with global invariant feature concatenation. Output features are passed through a 3-layer MLP to obtain the final segmentation labels. All network layers have latent dimension 128. For the neighborhood search in our message-passing layers, we use a ball query with radius $r = 0.3$ with maximum $k = 40$ points. We use an Adam optimizer with an initial learning rate of 0.001. All networks are trained on one single NVIDIA Titan RTX 24GB GPU.

## S.5 Additional Experiments

### S.5.1 Evaluation of Network Lipschitz (Sec. 5.3)

Here we plot the training-time Lipschitz regularization losses under $l_2$-norm in Fig. S1. The losses are applied to $\mathbf{y}$ samples both near the ground-truth fixed-point and uniformly distributed in the space. As shown in the plot, the losses are zero almost everywhere.

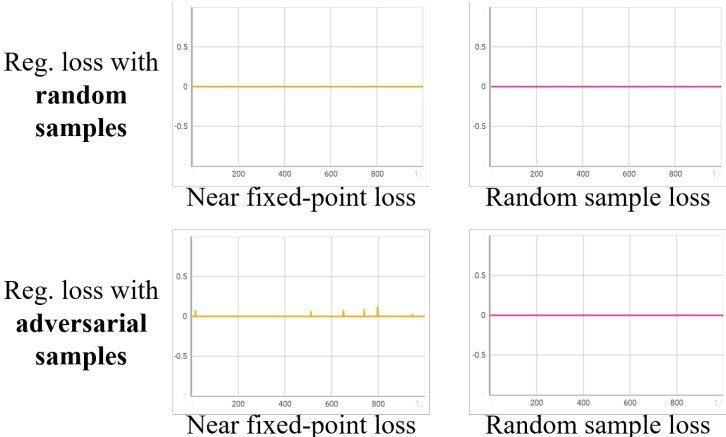

Figure S1: **Lipschitz regularization losses at training time.** We apply the regularization to $\mathbf{y}$ samples both near the ground-truth fixed-point (**left**) and uniformly distributed in the space (**right**).

### S.5.2 Model Convergence and Inference Time

In our experiments, we set $k = 20$ iterations for evaluation. But in practice, we plot the IoU w.r.t. the number of iterations in Fig. S2 and observe that the network prediction converges within $\sim k = 5$ iterations.

The training time is the same as the standard training frameworks as it only takes a single-step prediction with the ground-truth labels. If one wants to further incorporate Lipschitz regularization

losses into the training, more time is needed for the loss computation, especially for the adversarial sampling which involves the computation of network gradients and iterative sampling. But as stated in Sec. 4 and experimented in Sec. 5.3, we are not incorporating such regularizations in our current framework.

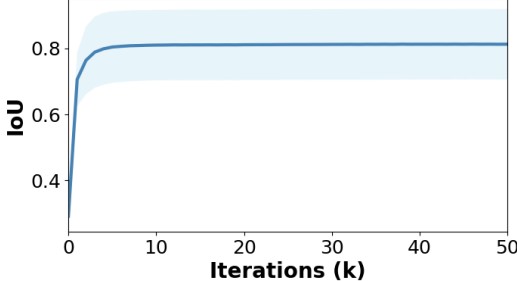

Figure S2: **IoU w.r.t. iterations.** On average, the network converges within $\sim k = 5$ iterations at inference time.

### S.5.3 Data Augmentation and Noise Stability

**Data augmentation on part poses**. We further compare our methods to the baselines under different data augmentation settings: no augmentation, global pose augmentation $\mathrm{SE}(3)$, and per-part pose augmentation $\mathrm{SE}(3)^P$. The results are in Tab. S1 below.

| Data aug. | **None** | $\mathrm{SE}(3)$ | $\mathrm{SE}(3)^P$ |
|---|---|---|---|
| PointNet | $46.15_{\pm 3.18}$ | $44.01_{\pm 2.43}$ | $43.07_{\pm 2.46}$ |
| DGCNN | $46.60_{\pm 4.03}$ | $37.18_{\pm 8.38}$ | $42.70_{\pm 4.22}$ |
| VNN | $47.09_{\pm 1.84}$ | $51.96_{\pm 8.11}$ | $46.29_{\pm 13.67}$ |
| Ours | $\mathbf{82.84}_{\pm 8.13}$ | $\mathbf{84.76}_{\pm 7.67}$ | $\mathbf{76.22}_{\pm 13.26}$ |

Table S1: **Data augmentation.** Networks are trained with no augmentation, global pose augmentation $(\mathrm{SE}(3))$, and per-part pose augmentation $(\mathrm{SE}(3)^P)$.

**Network stability to pointcloud noise**. We show the noise-stability analysis of our method in Fig. S3 below. We apply Gaussian noises to the test pointclouds with different standard deviations ranging from 0 to 0.05 and evaluate the networks trained on clean pointclouds and trained with augmentation of the same noise level as inference.

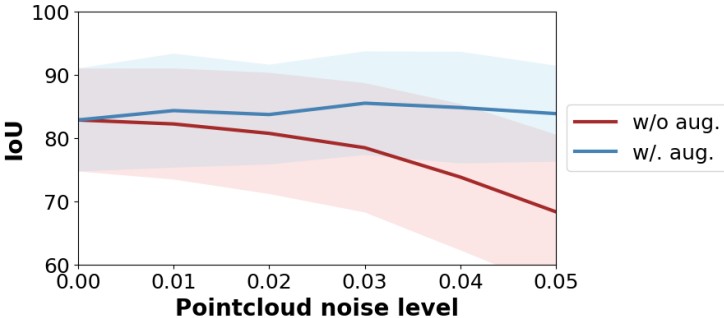

Figure S3: **Noise-stability.** We apply Gaussian noises to the input pointclouds with different standard deviations and test the networks trained on clean pointclouds (red) and trained with augmentation of the same noise level as inference (blue).