# OpenReview forum: "Banana: Banach Fixed-Point Network for Pointcloud Segmentation with Inter-Part Equivariance"
_NeurIPS.cc/2023/Conference — NeurIPS 2023 spotlight_

### Official Review · Reviewer_4UMu · 2023-06-14

**Soundness:** 4 excellent
**Presentation:** 4 excellent
**Contribution:** 3 good
**Rating:** 7
**Confidence:** 4

**Summary:**

This paper considers an important problem in learning on point clouds -- the equivariance under the SE(3) group.
Namely, the authors address the requirement of inter-part equivariance, essential for handling real-world scenarios, where an object can consist of multiple moving parts, or a scene can contain multiple objects that undergo different rigid transformations.

The key observation is that segmentation is necessary to define such per-part equivariance, which the authors are the first to do in a strict manner.
Furthermore, the proposed fixed-point framework with one-step training and iterative inference is used to demonstrate that the per-step equivariance induces an overall equivariance upon convergence.
The developed inter-part equivariant message-passing network with stable convergence is experimentally shown to have strong generalization under different scene configurations, even those changing the point cloud geometry/topology.

Overall, the paper provides a sound theoretical framework for a complex practical problem.

---------------------------------------
POST REBUTTAL

The authors comprehensively addressed my questions and concerns.
Given the provided answers and rebuttal overall, I maintained my positive assessment of the paper.

**Strengths:**

S1. The paper is very well-written, and the structure is sound. The illustrations are clear and instructive, which helps to comprehend the presented theory.



S2. The proposed view of inter-part equivariance as a co-evolving interplay between geometry and segmentation is novel and compelling.



S3. The experimental validation supports well the claims stated in the contribution.

**Weaknesses:**

W1. A discussion on the (time-)complexity of the proposed method is missing.

- It is unclear how efficient the iterative inference is.
- Besides, a comparison of the proposed model complexity and the baselines is missing.



W2. The details of the hyperparameter choice, including the iterative inference part, are missing.

- Crucially, the optimal number of iterations $\textbf{k}$ is unspecified, nor is its effect on the complexity with regards to W1.

- The same applies to the motivation of the chosen size of the radius of the ball query, $r$, and the maximum number of points in the local neighborhood, $k$.



W3. Experiment Section 5.1: the selected Shape2Motion shapes have at most 3 parts with two different semantic labels.

- I wonder how the proposed method would perform on Shape2Motion shapes with more than 2 different part labels, e.g., motorbike or bicycle.

**Questions:**

Q1. The first part of the title is often associated with negative connotations in English; see, e.g., [0]. The authors might want to revise it so as not to cause unnecessary misunderstanding.

Q2. How can the proposed method be used to perform point cloud classification?

Q3. The declaration of $φ$ in (14) is missing.

Q4. What is the number of runs in the experiment statistics presented in the tables in Section 5? Especially important for Table 2.

Q5. Can the authors explain the large standard deviation of the results in Table 1? Why does the proposed model, in most cases, perform better on Unseen states + unseen instances than Unseen states (see Table 1)?

Q6. Ablation studies: Was the inherent ambiguity of PCA-based canonicalization (see, e.g., [1]) taken care of? Could authors elaborate on the details of this experiment?

Q7. In the theory of the VN framework [2], which the authors' method is based on, the pose-effect cancelation is achieved by means of the inner product of the equivariant features, which also cancels the effect of reflections. Could the authors run a simple experiment showing if their method is actually E(3)-equivariant and discuss it?

Q8. Further noise-stability analysis of the method, where the noise is applied to the input point cloud (not just the segmentation mask), would be beneficial.

Q9. It would make the paper more accessible to a broader audience if the authors included an informal motivation of using the Banach fixed-point theorem and iterations.


A non-exhaustive list of typos:

- Line 148 " and we" --> ", we"

- Line 277 "espite" --> "despite"

-------------------------------------------------

[0]  "banana." Farlex Dictionary of Idioms. 2015. Farlex, Inc 14 Jun. 2023 https://idioms.thefreedictionary.com/banana



[1] Li et al. (2021), A Closer Look at Rotation-Invariant Deep Point Cloud Analysis



[2] Deng et al. (2021), Vector neurons: A general framework for SO(3)-equivariant networks

**Limitations:**

The authors adequately addressed the limitations and broader societal impact.

---

> ### Author Rebuttal · Authors · 2023-08-10
>
>
> Thanks for your feedback and for appreciating our work! Here’re our responses to your questions and comments and we hope that they could address your concerns as well:
>
> **W1. Time complexity of the proposed method is missing.**
>
> In our experiments, we set k=20 iterations for evaluation. But in practice, we plot the IoU w.r.t. the number of iterations (PDF Figure 2) and observe that the network prediction converges within ~k=5 iterations. We will add this information to our paper.
>
> The training time is the same as the standard training frameworks as it only takes a single-step prediction with the ground-truth labels. If one wants to further incorporate Lipschitz regularization losses into the training, it will take more time for the loss computation, especially if it uses adversarial sampling which needs to compute the network gradients and sample iterations. But currently we are not incorporating such regularizations.
>
> **W2. Details of the hyperparameter choice.**
> As mentioned above, for the experiments in the main paper, we set k=20 iterations for inference, but on average the network converges within ~k=5 iterations.
>
> The input pointcloud contains 2048 points and is scaled to [-1, 1]. For the message passing layers, we set a neighborhood radius r=0.3 with maximum number of 40 points. An ablation study on the radius is shown in Table 4 of the attached PDF. When the radius is too small, the network fails to extract useful local features; and when the radius is 0.3~0.4 the network performance is relatively consistent.
>
> We will add these details to the paper.
>
> **W3. Experiment Section 5.1: the selected Shape2Motion shapes have at most 3 parts with two different semantic labels.**
>
> We demonstrate our method on these object categories mostly because their part motions are more obvious, which helps us to show our “inter-part equivariance”. For categories like motorbike or bicycle, the part motions are relatively small (e.g. wheels rotating). We believe for several parts, our method can also work well. But if it's tens of parts like scene segmentation, the dimension of the iteration space $[0,1]^P$ greatly increases and it may cause difficulties.
>
> **Q1. Acronym “Banana”.**
>
> In fact, the acronym “Banana” comes from a meme in some math departments where students jokingly call “Banach space” the “Banana space”. But we are very sorry for not being aware of its negative connotations in English and we will consider changing or removing it.
>
> **Q2. How can the proposed method be used to perform point cloud classification?**
>
> As the fixed-point iteration must be always in the same domain $[0, 1]^P$, the direct input of our framework can only be the segmentation labels. However, if we don’t restrict the whole pipeline to be end-to-end, one can first do segmentation and apply another standard part-aware equivariant network to do other tasks like classification.
>
> **Q3. The declaration of $\varphi$ in (14) is missing.**
>
> Thanks for pointing it out! $\varphi$ is an MLP representing the edge function in the message passing. We will add the declaration to the paper.
>
> **Q4. Number of runs.**
>
> The error bars are not w.r.t. different random seeds, but are the standard deviation across different instances in the datasets.
>
> **Q5. The large standard deviation in Table 1. Better results on Unseen states + unseen instances than Unseen states.**
>
> For the large standard deviation, we notice that unlike many standard segmentation networks where prediction errors often exist locally (e.g. mislabeling a small region), our errors usually happen in a way that it converges to a totally different fixed point, e.g. an oven with its door slightly open, the network converges to a state where another wall of the oven body is labeled as “door” and the door is labeled as “body”. In other words, our errors usually exist globally instead of locally.
>
> The better performances on unseen instances seem to also happen with all the baseline methods. Our guess is that it may be due to the shape biases between the two sets. For example, in the washing machine category, the majority of the instances have oval doors and the minority have rectangular doors, and there’re more rectangular doors in the test split than in the training split.
>
> **Q6. Inherent ambiguity of PCA-based canonicalization.**
>
> Yes, the inherent ambiguity of PCA is eliminated in the canonicalization with sign-flipping based on the mean at each PCA direction. The infinite Lipschitz upper bound of PCA comes from the possibility that a small change in the part assignment $\mathbf{y}$ can result in a substantial change in the PCA directions. For example, imagine you have a part whose shape is a perfect sphere, now if any point around this part is added to this part (with weight $\mathbf{y}_n = \varepsilon$), the point’s direction will immediately become a principle direction of the part – such direction change caused by one single point label change is an example of unbounded Lipschitz.
>
> **Q7. E(3)-equivariance.**
>
> Yes, the VN framework we adopt is indeed E(3)-equivariant. A sketchy draft discussing the reflection-equivariance and potentially how to break such symmetry can be found at: https://arxiv.org/pdf/2210.16646.pdf
>
> **Q8. Noise-stability analysis of the method.**
>
> We show the noise-stability analysis of our method in PDF Figure 4. We apply Gaussian noises to the input pointclouds with different standard deviations ranging from 0 to 0.05 (the value 0.05 is adopted from [1]). In Figure 6 of the main paper, we also show some qualitative results of our method trained on clean synthetic pointclouds and tested on noisy real scans.
>
> [1] Mescheder, Lars, et al. "Occupancy networks: Learning 3d reconstruction in function space." Proceedings of the IEEE/CVF conference on computer vision and pattern recognition. 2019.
>
> **Q9. Informal motivation for using the Banach fixed-point theorem and iterations.**
>
> Thanks for the suggestion! We will add some motivations to the paper.

---

> > ### Comment · Reviewer_4UMu · 2023-08-11
> >
> > Thank you for comprehensively answering my questions and addressing my concerns!
> >
> > I find your additional experimental results convincing and recommend you include them in the paper.
> > Given the provided answers and rebuttal overall, I intend to maintain my positive assessment of the paper.

---

> > > ### Author Response · Authors · 2023-08-21
> > >
> > > Thanks for your reply! We'll add the additional results to the paper/supplementary material.

---

### Official Review · Reviewer_T7gc · 2023-06-24

**Soundness:** 3 good
**Presentation:** 4 excellent
**Contribution:** 3 good
**Rating:** 6
**Confidence:** 4

**Summary:**

The authors introduce a method for object part segmentation of point clouds that is equivariant/invariant to SE(3) part transformations. The core of their method is a neural network with points and segmentation as input and segmentation as output. The network is assumed to be contractive and is then used to perform Banach fixed point iterations towards the correct segmentation during inference. The Banach fixed point network is embedded in a larger architecture using Vector Neurons. For implementation, a message passing network is used, which uses the input segmentation for determining message strength.

The proposed method is evaluated for articulated object part segmentation (on Shape2Motion) and for segmentation of multi-object scans (on DynLab), where it compares favorably against baselines.

**Strengths:**

- The theoretical framework of Banach iterations is elegant and an interesting perspective on iterative networks during inference. In practice, similar ideas on point clouds have been there before, understanding such networks as reweighting functions for iterative reweighting least square schemes, EM iterations, etc. However, I think there is value in this specific perspective, especially since it comes with a convergence guarantee under Lipschitz constraints. To my knowledge, this is a novel contribution.
- The presented algorithm seems to solve the given tasks well, outperforming the given baselines clearly (however, there are some concerns regarding missing baselines, see below).
- The paper is nicely presented and easy to follow, given the slightly more complex nature.
- The result of generalization from synthetic to real chairs is strong

**Weaknesses:**

- The first part of the paper is all about the consequences of a Lipschitz-constraint network and then there is no Lipschitz constraint in practice. The authors claim that the local message passing serves as some type of Lipschitz regularization. However, there is no reference given for this claim and I don't know one either.  I think it would make the paper stronger if this aspect would be supported by experiments or theory, which is isn't.  The authors only ablate on model performance under different noise levels, which does not give a full picture about convergence.
- In articulated object part segmentation,  the paper seems to leave out important comparisons to previous work, e.g., [1,2,3]. All three papers are referenced in related work but not compared against. It would be good if the authors would provide a comparison or a discussion, why the comparison is not necessary.

I think this is a very interesting paper with a new perspective on iterative networks. However, the above points are a bit concerning, which is why I am not fully convinced to give a better score.

[1] Kawana et al.: Unsupervised pose-aware part decomposition for 3d articulated objects.

[2] Kawana et al.: Neural star domain as primitive representation

[3] Chen et al.: BAE-NET: Branched Autoencoder for Shape Co-Segmentation

**Questions:**

- I suspect that equation 1 wants to express linear blending of part transforms. However, I don't think it is correct: (1) sum of matrix $\mathbf{R}$ and vector \mathbf{t}, (2) $\mathbf{t}_p$ not weighted by $\mathbf{y}_{np}$ . Could the authors clarify?
- Please provide further evidence that the network indeed behaves contractive or a theoretical justification for the Lipschitz regularization via local message passing.
- Please discuss or provide missing comparisons with previous methods for part segmentation on Shape2Motion.

**Limitations:**

The authors discuss limitations and societal impact.

---

> ### Author Rebuttal · Authors · 2023-08-09
>
> Thanks for your feedback and for appreciating our work! Here’re our responses to your questions and comments and we hope that they could address your concerns as well:
>
> **I suspect that equation 1 wants to express linear blending of part transforms. However, I don't think it is correct: Could the authors clarify?**
>
> Yes, it is a typo. Thanks for pointing it out! It should be $\sum_{p=1}^P \mathbf{y}_{np}(\mathbf{x}_n\mathbf{R}_p+\mathbf{t}_p)$. We made this mistake because we were writing $(\mathbf{R}_p, \mathbf{t}_p)$ together as $\mathbf{T}_p$ in the beginning and changed it in the last minute but forgot to change the expressions.
>
> **Please provide further evidence that the network indeed behaves contractive or a theoretical justification for the Lipschitz regularization via local message passing.**
>
> We have added a paragraph of (informal) theoretical explanation of why the SE(3)-equivariant message passing is helpful to small network Lipschitz constants. The PDF file is directly sent to AC as it contains TeX equations that cannot be rendered by MathJax here in the text box.
>
>
> In addition, the convergence itself actually doesn’t rely on the network being contractive with $L<1$, as $[0,1]^P$ is a compact convex space and the Brouwer fixed-point theorem guarantees the existence of fixed-points for any continuous functions, which can be found by Newton fixed-point iterations. However, we developed our theory on top of the Banach fixed-point theorem/iteration for the uniqueness of the fixed point, which is for the proof of equivariance.
>
>
> We also provide a study of different Lipschitz constraining methods under different norms (PDF Table 1 and Figure 1). In practice, none of these existing Lipschitz constraining methods are helpful to the network performance (PDF Table 1). We also plot the $l_2$-norm regularization losses (PDF Figure 1) and they are zero almost everywhere.
>
>
> Directly evaluating the network Lipschitz is also non-practical, as the space $[0, 1]^P$ is of very high dimension and the sampling in it is very inefficient. This inefficient sampling also makes the behavior of the Lipschitz regularization losses less understandable and controllable, as they’re not regularizing the entire space but only on some sparsely sampled points, which may bring unexpected natures to the space.
>
> Overall, based on our observations, not having an explicit Lipschitz constraint and using the SE(3)-equivariant message passing work best for the current situation. But we also agree that, as we discussed in our limitation section, Lipschitz bounds and regularizations for set-invariant networks (which norm to use and how to constrain) would be an interesting and important problem for future study.
>
> **Please discuss or provide missing comparisons with previous methods for part segmentation on Shape2Motion.**
>
> The assumptions and training/test setups of [1, 2, 3] are different from ours in the following aspects:
> -  [1, 2, 3] are unsupervised **co-segmentation**, ([1] uses a GAN loss, and [2, 3] follows the slot-attention-like techniques), but we are following the standard supervised training/test frameworks for semantic/part segmentations.
> - Most importantly, [1, 2, 3] require the training data to have objects at **all articulation states**, but we train our method only on limited states (e.g. only the rest state) and show its generalization to unseen states.
> - Another minor point is that [1, 2, 3] also needs watertight meshes/implicits for their training, but our method can work on pure pointcloud data.
>
> [2, 3] doesn’t apply directly to articulated objects, but similar ideas are incorporated in [4] with part-level SE(3)-equivariance for articulated object segmentation (also discussed in the related work section) – and the two arguments above hold for [4] as well. Also note that although [4] shows that they feed their segmentation and pose predictions back to their feature extractor in their pipeline figure, they only do a two-step coarse-to-fine prediction and are not actually using an iterative framework – and also they only leverage per-part SE(3)-equivarinace but don’t provide any theoretical insights for inter-part equivariance.
>
> We will add these discussions to our paper.
>
> [1] Kawana et al.: Unsupervised pose-aware part decomposition for 3d articulated objects.
>
> [2] Kawana et al.: Neural star domain as primitive representation
>
> [3] Chen et al.: BAE-NET: Branched Autoencoder for Shape Co-Segmentation
>
> [4] Liu, Xueyi, et al. "Self-Supervised Category-Level Articulated Object Pose Estimation with Part-Level SE (3) Equivariance." arXiv preprint arXiv:2302.14268 (2023).

---

> > ### Comment · Reviewer_T7gc · 2023-08-20
> > **Thanks**
> >
> > Thank you for explaining the differences in setup with respect to related works [1 - 4]. I would encourage to make this distinction also clear in the related work section of the paper. I guess one could still compare the methods in a setup where all articulation states are used as training data but I agree that this might not be essential to support the main argument of the paper.
> >
> > I appreciate the theoretical justification for message passing networks. It provides a good framework of how to think about it. Also, the additional results shown in the PDF are good additions to the paper. One concern I have regarding the Lipschitz losses: If they are zero everywhere when applied as additional loss, doesn't this just mean they are weighted to strongly? If this is the case it is also no wonder that they have strong negative impact on the IoU. Wouldn't it be more interesting to show these plots in a scenario where they are not used in the loss in order to show the behaviour of the network without additional regularization?
> >
> > All in all, I am satisfied with the answers to my question and increase my score. I think even if this method is not directly relevant in practice, the theoretical framework is interesting and investigates iterative networks from a new perspective.

---

> > > ### Author Response · Authors · 2023-08-21
> > >
> > > Thanks for your reply and for raising your score! We'll add the additional discussions and results to the paper/supplementary material.

---

### Official Review · Reviewer_tamA · 2023-06-26

**Soundness:** 4 excellent
**Presentation:** 4 excellent
**Contribution:** 4 excellent
**Rating:** 8
**Confidence:** 4

**Summary:**

The paper proposes an equivariant network for part-based (or multi-object) point cloud segmentation. The approach is equivariant to separate SE(3) transformations of each part/object. This is ensured by introducing a Banach fixed-point network. The network takes the point-could and the current segmentation as input, and iterates till convergence. The proposed part-aware equivariant network employs the Vector Neurons method to achieve equivariance for the individual parts. A segmentation-weighted message passing then adds communication between the different parts. Experiments are performed on objects from the Shape2Motion and DynLab datasets.

**Strengths:**

-	Novel work based on very interesting and elegant ideas.
-	Theoretically sound.
-	Very well written and clear.
-	Good illustrations.
-	Comparison with several methods.


**Weaknesses:**

1.	The experimental evaluation is limited to very small-scale datasets. It is not clear how the method would scale to larger datasets and more complex scenes, e.g. in an automotive setting, or from terrestrial Lidar scans. It would be good if the authors could discuss this in more detail.

2.	It seems that the authors always train on the exact instance which is also encountered during inference. How would the method perform if it encounters a new type of, e.g., oven, after being trained on a dataset of different ovens.

3.	I did not find how the other methods in table 1 and 2 were trained. Is data augmentation used? How would data augmentation impact the performance of e.g. PintNet++ or MeteorNet?

4.	I did not find discussion on inference and training time.

5.	It would be very interesting to see some plots of the IoU w.r.t. the number of iterations, to understand the convergence behavior of the model.


In summary, I think that this is a very interesting and solid work. I find no significant weaknesses. Although it would be appreciated if the authors can answer and address my comments in a rebuttal to further strengthen the paper.

**Questions:**

See weaknesses.

Moreover:
Which other tasks could be suitable for this method?

**Limitations:**

They are well addressed.

---

> ### Author Rebuttal · Authors · 2023-08-09
>
> Thanks for your feedback and for appreciating our work! Here’re our responses to your questions and comments and we hope that they could address your concerns as well:
>
> **1. Generalization to larger datasets and more complex scenes.**
>
> For larger scenes, we believe a major difficulty would be the network architecture. Currently, we are using the Vector Neuron framework for equivariance and the message passing for encouraging small network Lipschitz. But in larger scenes, more complex network structures are usually important, e.g. transformers, down/up-sampling, voting schemes, and how to incorporate equivariance and Lipschitz properties into these complex networks is yet underexplored.
>
> **2. Inference on novel instances.**
>
> In Table 1 right and Figure 5 right in the main paper, our method is tested on novel instances which are not seen in the training set. We also show in Figure 6 (main paper) that after training on the clean synthetic samples, our method can be applied to real scans with some noise. We visualize more paired training and test instances of different states only for a better illustration of the equivariance properties.
>
> **3. Data augmentation.**
>
> We further compare our methods to the baselines under different data augmentation settings: no augmentation, global pose augmentation $SE(3)$, and per-part pose augmentation $SE(3)^P$. The results are in Table 3 in the attached PDF file.
>
> **4. Inference and training time. 5. Convergence behavior of the model.**
>
> In our experiments, we set k=20 iterations for evaluation. But in practice, we plot the IoU w.r.t. the number of iterations (PDF Figure 2) and observe that the network prediction converges within ~k=5 iterations. We will add this information to our paper.
>
> The training time is the same as the standard training frameworks as it only takes a single-step prediction with the ground-truth labels. If one wants to further incorporate Lipschitz regularization losses into the training, it will take more time for the loss computation, especially if it uses adversarial sampling which needs to compute the network gradients and sample iterations. But currently, we are not incorporating such regularizations.
>
> **Moreover: Which other tasks could be suitable for this method?**
>
> We think one task that may be very suitable for our method is tracking. As the changes between consecutive frames are small in tracking, one can probably use the segmentations from timestep $t$ as an initialization for the iterations at timestep $t+1$, which might be very helpful for the convergence. And the equivariance properties, on the other hand, may help the tracking network achieve better inter-frame consistency.

---

> > ### Comment · Reviewer_tamA · 2023-08-13
> >
> > I thank the authors for the addressing my questions. I will maintain my positive rating.

---

> > > ### Author Response · Authors · 2023-08-21
> > >
> > > Thanks for your reply!

---

### Official Review · Reviewer_DYXN · 2023-06-28

**Soundness:** 3 good
**Presentation:** 3 good
**Contribution:** 2 fair
**Rating:** 7
**Confidence:** 1

**Summary:**

The paper propose a Banach, an approach for part-based point-cloud segmentation. In particular, the authors propose an approach to enforce equivariance of the part-segmentation, by construction.  They propose a fixed-point framework with one-step training and iterative inference. They propose a part-aware segmentation network.

**Strengths:**

- The results are very convincing - the proposed approach seem to significantly outperform previous approaches

**Weaknesses:**

I do not see any direct weaknesses but I have very little knowledge about this field.

**Questions:**

- what is the run-time of the approach? How does it compare to previous approaches?

**Limitations:**

yes

---

> ### Author Rebuttal · Authors · 2023-08-09
>
> Thanks for your feedback and for appreciating our work! Here’re our responses to your questions and comments and we hope that they could address your concerns as well:
>
> **Inference and training time.**
>
> In our experiments, we set k=20 iterations for evaluation. But in practice, we plot the IoU w.r.t. the number of iterations (PDF Figure 2) and observe that the network prediction converges within ~k=5 iterations. We will add these information to our paper.
>
> The training time is the same as the standard training frameworks as it only takes a single-step prediction with the ground-truth labels. If one wants to further incorporate Lipschitz regularization losses into the training, it will take more time for the loss computation, especially if it uses adversarial sampling which needs to compute the network gradients and sample iterations. But currently we are not incorporating such regularizations.

---

> ### Comment · Reviewer_DYXN · 2023-08-16
>
> Since I had no major concerns, I will update my score to 7

---

> > ### Author Response · Authors · 2023-08-21
> >
> > Thanks for your reply and for raising your score!

---

### Official Review · Reviewer_iMm7 · 2023-07-06

**Soundness:** 3 good
**Presentation:** 3 good
**Contribution:** 3 good
**Rating:** 6
**Confidence:** 3

**Summary:**

The paper proposes a neural network architecture that is equivariant to transformations in SE(3) for each object part independently. The network is part of a fixed-point framework where the network is trained with a single step but during testing an iterative approach is used that converges to the desired segmentation.

**Strengths:**

- The paper presents a fresh idea to solve the problem of part segmentation that takes into consideration the different movable parts that compose an object and makes the network equivariant to transformations of those.

- The paper provides theoretical derivations that motivate the solution proposed in the paper.

- The paper shows the effectiveness of the solution by providing several experiments.

**Weaknesses:**

Although I like the paper, I believe the evaluation could be improved by including other types of equivariant networks:

- For example, neural networks based on equivariant operations such as group convolutions or steerable convolutions have by construction equivariance wrt the object parts too. Although these methods would allow information flow between object parts, most of the object parts would remain equivariant to transformations of those.

- Moreover, network architectures that work only with the intrinsic information of the shape should also be included. Graph Convolution networks or equivariant mesh convolutions would also maintain equivariance of object parts. These networks are commonly used to segment people in different poses, which is a related problem to the one addressed in the paper.

- As the paper states, there is no guarantee that L < 1. The paper states that weight truncation could restrict the upper bound of L but harm the expressivity of the network. An experiment where this is studied would improve the paper.

- Lastly, I do not like the acronym Banana since it is not really an acronym of the title. There is no need to include an acronym on the paper.

**Questions:**

See the weaknesses section.

**Limitations:**

The limitations are well addressed.

---

> ### Author Rebuttal · Authors · 2023-08-09
>
> Thanks for your feedback and for appreciating our work! Here’re our responses to your questions and comments and we hope that they could address your concerns as well:
>
> **Equivariant convolutions.**
>
> The VNN baseline we compare to is also using a graph convolution network backbone. These types of methods face the dilemma of local neighborhood size – when the neighborhoods are too small, it’s more agnostic of part motions but the local features are less expressive, and the contrary when the neighborhoods are large. We also add a baseline comparison with invariant-feature message passing (PDF Table 2, also discussed below), which is an extreme case where local equivariant features are only extracted in the first layer.
>
> **Intrinsic methods.**
>
> We agree that intrinsic methods are highly relevant to our task and will add discussions of these works to the paper. Compare to intrinsic methods, our method has the following superiorities:
> - Rigid part motions preserve geometric distances in many cases, however, this doesn’t hold when topological changes exist, e.g. oven door closed to oven door opened, where the intrinsic operators such as the Laplacian cannot stay constant (PDF Figure 3). And most intrinsic methods (either spectral methods like the functional map, or spatial methods like the graph convolution) are developed based on the Laplacian operator.
> - Another limitation of intrinsic methods is that they cannot do part segmentation with duplicated parts (like the chairs example in Figure 6, main paper), as the same local geometries can only be classified as the same labels. In other words, our method can give “equivariant” (order 1) outputs, but intrinsic methods can only give “invariant” (order 0) outputs.
> - Also, implementing intrinsic operators on pointclouds is much more non-trivial than on meshes, especially when noises are also considered.
>
> In addition, we also add a baseline network which first computes per-point local SE(3)-equivariant features, converts them to invariant features, and applies invariant message passing (PDF Table 2), which is a strategy adopted for human segmentation in [1].
>
> [1] Feng, Haiwen, et al. "Generalizing Neural Human Fitting to Unseen Poses With Articulated SE (3) Equivariance." arXiv preprint arXiv:2304.10528 (2023).
>
> **Network Lipschitz.**
>
> We have added a paragraph of (informal) theoretical explanation of why the SE(3)-equivariant message passing is helpful to small network Lipschitz constants. The PDF file is directly sent to AC as it contains TeX equations that cannot be rendered by MathJax here in the text box.
>
>
> In addition, the convergence itself actually doesn’t rely on the network being contractive with $L<1$, as $[0,1]^P$ is a compact convex space and the Brouwer fixed-point theorem guarantees the existence of fixed-points for any continuous functions, which can be found by Newton fixed-point iterations. However, we developed our theory on top of the Banach fixed-point theorem/iteration for the uniqueness of the fixed point, which is for the proof of equivariance.
>
>
> We also provide a study of different Lipschitz constraining methods under different norms (PDF Table 1 and Figure 1). In practice, none of these existing Lipschitz constraining methods are helpful to the network performance (PDF Table 1). We also plot the $l_2$-norm regularization losses (PDF Figure 1) and they are zero almost everywhere.
>
>
> Directly evaluating the network Lipschitz is also non-practical, as the space $[0, 1]^P$ is of very high dimension and the sampling in it is very inefficient. This inefficient sampling also makes the behavior of the Lipschitz regularization losses less understandable and controllable, as they’re not regularizing the entire space but only on some sparsely sampled points, which may bring unexpected natures to the space.
>
> Overall, based on our observations, not having an explicit Lipschitz constraint and using the SE(3)-equivariant message passing work best for the current situation. But we also agree that, as we discussed in our limitation section, Lipschitz bounds and regularizations for set-invariant networks (which norm to use and how to constrain) would be an interesting and important problem for future study.
>
> **Acronym “Banana”.**
>
> In fact, the acronym “Banana” comes from a meme in some math departments where students jokingly call “Banach space” the “Banana space”. But we are very sorry for not being aware of its negative connotations in English and we will consider changing or removing it.

---

> > ### Comment · Reviewer_iMm7 · 2023-08-16
> > **Answer to rebuttal**
> >
> > Thank you for the thorough rebuttal. All my concerns have been addressed and I would encourage the authors to include these discussions in the paper and/or supplementary material. Regarding the acronym, I was not aware of this since mathematics is not my background. I would not oppose leaving it in the paper. I keep my initial positive assessment of the paper.

---

> > > ### Author Response · Authors · 2023-08-16
> > >
> > > Thanks for your reply!
> > >
> > > And thanks for your understanding of the paper title. In fact, we also have a backup plan to call it "La Banane" ("The Banana" in French), which is the acronym for **Bana**ch **N**etwork with **E**quivariance (an actual acronym!), and it won't have negative connotations in English as it is not English... We used "Banana" in this submission because we felt having a French word in the title may make it look a bit obscure.
> > >
> > > And yes, we'll add the contents to the paper/supplementary material!

---

### Author Rebuttal · Authors · 2023-08-09

We thank all reviewers for their constructive feedbacks, and are glad that they find our work presenting novel and compelling ideas as well as convincing results.

Here we provide a brief summary of our response, including additional theoretical explanations and experimental evaluations. Detailed responses are replied to each reviewer separately. We attach a PDF file presenting tables/figures with brief explanations (for more detailed discussions please refer to our text response), and a PDF file directly sent to AC containing the equations that cannot be rendered in the text boxes. We will also add these contents as well as fix the typos in our paper revision.

- Network Lipschitz (iMm7, T7gc)
  - Theoretical explanations. (The file is directly sent to AC through an anonymous link as it contains equations that cannot be rendered by MathJax directly in the text boxes.)
  - Network performance under different Lipschitz constraining methods. [PDF file Table 1 and Figure 1]

- Algorithm complexity (tamA, DYXN)
  - Training and inference time compared to other methods (tamA, DYXN).
  - IoU w.r.t. the number of iterations (tamA). [PDF file Figure 2]

- Evaluations and comparisons (iMm7, tamA, T7gc, 4Umu)
  - Hyperparameters for neighborhood radius (4Umu). [PDF file Table 4]
  - Discussions of equivariant convolutions. Comparison to an invariant message passing strategy adopted for human segmentation (iMm7). [PDF file Table 2]
  - Discussions of intrinsic methods (iMm7). [PDF file Figure 3]
  - Data augmentation for the baseline methods (tamA). [PDF file Table 3]
  - Network stability w.r.t. different noise levels on the input pointclouds (4Umu). [PDF file Figure 4]
  - Discussions of other articulated-object segmentation methods (T7gc).

- Other discussions, including answers to questions, clarifications, and the acronym of the paper title (iMm7, tamA, T7gc, 4Umu)

---

### Decision · Program_Chairs · 2023-09-21

**Decision:**

Accept (spotlight)

**Comment:**

This paper addresses an important problem of learning equivariance for complex systems such as articulated objects or multi-object scenes. It presents an equivariance formulation that allows for their co-evolution, and proposes a Banach fixed-point network for equivariant segmentation with inter-part equivariance by construction. The paper also gives theoretical results for per-step equivariance and global convergence. Experiments verify the proposed method. Reviewers suggest inclusion of other types of equivariant networks and more technical discussions. It seems that the proposed method has the challenge of being used by other architectures for large-scale learning. The authors are suggested to include these additional discussions and results in the final version. Congratulations!